# Extending Power of Nature from Binary to Real-Valued Graph Learning in Real World

**Chunshu Wu**[1], **Ruibing Song**[1], **Chuan Liu**[1], **Yunan Yang**[2], **Ang Li**[3], **Michael Huang**[1], **Tong Geng**[1]

Department of Electrical and Computer Engineering, University of Rochester[1]
Department of Mathematics, Cornell University[2],
Physical & Computational Sciences Directorate, Pacific Northwestern National Laboratory[3]

## Abstract

Nature performs complex computations constantly at clearly lower cost and higher performance than digital computers. It is crucial to understand how to harness the unique computational power of nature in Machine Learning (ML). In the past decade, besides the development of Neural Networks (NNs), the community has also relentlessly explored nature-powered ML paradigms. Although most of them are still predominantly theoretical, a new practical paradigm enabled by the recent advent of CMOS-compatible room-temperature nature-based computers has emerged. By harnessing a dynamical system's intrinsic behavior of chasing the lowest energy state, this paradigm can solve some simple binary problems delivering considerable speedup and energy savings compared with NNs, while maintaining comparable accuracy. Regrettably, its values to the real world are highly constrained by its binary nature. A clear pathway to its extension to real-valued problems remains elusive. This paper aims to unleash this pathway by proposing a novel end-to-end Nature-Powered Graph Learning (NP-GL) framework. Specifically, through a three-dimensional co-design, NP-GL can leverage the spontaneous energy decrease in nature to efficiently solve real-valued graph learning problems. Experimental results across 4 real-world applications with 6 datasets demonstrate that NP-GL delivers, on average, $6.97 \times 10^3$ speedup and $10^5$ energy consumption reduction with comparable or even higher accuracy than Graph Neural Networks (GNNs).

## 1 Introduction

Nature apparently performs complex computations constantly, e.g., solving differential equations and performing random sampling, at clearly lower cost and higher performance than digital computers. Nature's computation can be powered by various phenomena in physics: entanglement and tunneling that drive quantum computing, the spontaneous energy decrease observed in dynamical systems that drives energy-based computing, and so on.

The latter happens ubiquitously in daily life – actually, many dynamical systems in nature spontaneously and swiftly evolve towards the most stable states with the lowest energy, during which highly complex computation is performed. Intuitive examples are ink diffusion in water and chemical reactions among molecules. These natural processes are extremely complex – to be evaluated with acceptable accuracy on digital computers, chemical reactions, such as drastic combustion processes involving only 100K atoms, can be formulated as over $10^7$ iterations of computation with each iteration requiring $10^9$ FLOPs (Wu et al., 2023). In contrast, nature could deliver accurate solutions to such complex problems within microseconds. That being said, if we can build computers that can efficiently harness the power of nature, these computers, so-called "***nature-based computers***", have the potential to solve complex real-world learning problems delivering substantial speedups and competitive accuracy compared with traditional solutions e.g., NNs on GPUs.

Intrigued by the unique computational power in nature, in the past decade, besides the tireless efforts on the development of NNs, the community has also relentlessly explored ML paradigms that are powered by natural principles and can be realized on nature-based computers. Typical examples are quantum ML (King et al., 2022) on quantum computers and optical ML on photonic or optical

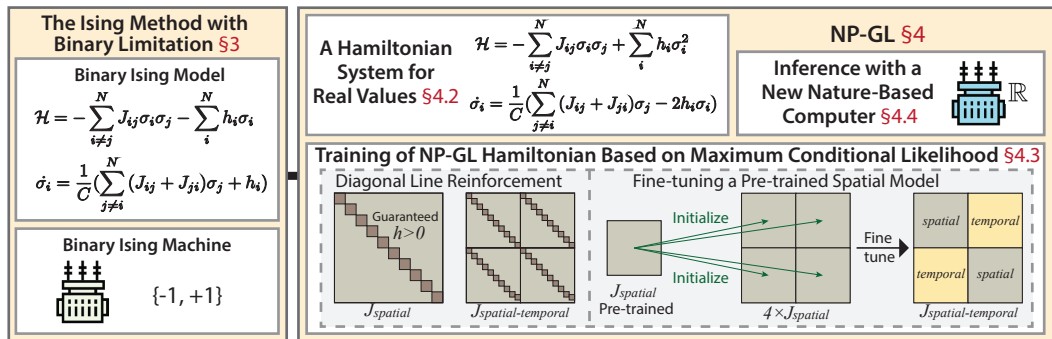

Figure 1: Overview of the end-to-end NP-GL framework.

systems (Inagaki et al., 2016b; Lin et al., 2018). Regrettably, despite their perceived potential, many nature-powered ML methods are still predominantly theoretical and demand stringent operational conditions. Notably, the emergence of the "Ising machine" (Afoakwa et al., 2021; Sharma et al., 2022) is closing the gap between theoretical nature-powered methodologies and practical applications. This novel computing paradigm, compatible with CMOS technology, can operate at room temperature with less than 1 Watt of power consumption, signifying a major leap towards unleashing the computational power of nature into the real world.

Specifically, Ising machines as a physical embodiment of the Ising model can be thought of as a dynamical system governed by the Hamiltonian (energy function of a dynamical system) of the Ising model[1] (detailed in Section 2.1). Driven by the spontaneous energy decrease in nature, a CMOS-based Ising machine can swiftly and automatically chase and find its lowest-energy states at the "speed of electrons" with negligible costs – $mW$ power & $ns$ latency. Unsurprisingly, the unique computational power of Ising machines has enabled a new nature-powered ML paradigm – termed "***Ising Graph Learning (IGL)***" in this paper. In IGL, a graph problem (e.g., time series forecasting) is formulated as an Ising model whose parameters are trained to map the problem's desired (e.g., prediction) results with higher probabilities to the lower energy states of the Ising machine, enabling the Ising machine to automatically find the desired solution at extreme speed (detailed in Section 2.2). It has been demonstrated that IGL outperforms GNNs with orders of magnitude speedups maintaining competitive accuracy in simple real-world graph learning applications with binary data, e.g., traffic congestion prediction (Pan et al., 2023) and collaborative filtering (Liu et al., 2023).

Despite the early-stage successes, a significant limitation persists within Ising Graph Learning: it only works with binary problems. The limitation is rooted in the definition of the Ising model – a node (aka "spin") of the Ising model has only two states. As the physical implementation of the Ising model, Ising machines are also designed for *binary* objects. Regrettably, as most critical applications in the real world use real-valued data, the practical values of IGL remain highly limited. If the notable advantages of Ising Graph Learning, driven by the automatic energy decrease, can be extended to the real number field, we foresee the ensuing nature-powered graph learning paradigm offering orders of magnitude speedups and energy savings compared to GNNs while preserving highly competitive accuracy for graph learning problems. A route to this extension remains elusive, and a clear pathway is highly desired.

To this end, this paper proposes an end-to-end Nature-Powered Graph Learning (NP-GL) framework, extending nature's computing power inherent in dynamical systems to tackle real-valued and real-world graph learning problems. The framework's overview, encompassing four components, is depicted in Figure 1: ***Limitation analysis of Ising Graph Learning:*** We theoretically analyze why the vanilla Ising Graph Learning (including both the Ising model and Ising machine) cannot be straightforwardly extended to support real values without upgrading the Ising Hamiltonian. ***Exploration of NP-GL Hamiltonian:*** Based on the analysis, NP-GL incorporates a newly designed Hamiltonian that is highly hardware-friendly. It inherits the strengths of the Ising model, ensuring high expressivity, and maintains distinct stable states with real values. ***Design of NP-GL training algorithms:*** Similarly to Ising Graph Learning, the parameters of the new Hamiltonian are trained to construct an energy landscape, in which the lowest-energy states correspond to the ground truth derived from historical data. In pursuit of high accuracy, NP-GL training adopts an improved conditional likelihood method with two optimizations: (1) Diagonal Line Reinforcement, which re-

---

[1]The Ising model is a probabilistic graphical model widely used to represent binary-state dynamical systems.

inforces the self-reaction parameters for better temporal continuity and differentiates the training of self-reaction (diagonal) and coupling parameters that convey distinct physical information. (2) Spatial Pre-training coupled with Temporal Fine-tuning, which leverages the similarity between spatial and temporal correlations to better learn from the temporal information. ***Design of NP-GL nature-based computer:*** Due to the similarity between NP-GL and Ising Hamiltonians, we build the new nature-based computer governed by the NP-GL Hamiltonian by slightly augmenting the circuitry of the Ising machine. The new nature-based computer, like Ising machines, leverages nature's power to swiftly find the lowest-energy states, but, unlike them, it does so with real values.

To the best of our knowledge, NP-GL is the first end-to-end real-valued nature-powered ML solution that outperforms NNs in the real world. Our contributions are summarized as follows.

- We propose NP-GL, an end-to-end nature-powered graph learning method, through codesign of Hamiltonian, training algorithms, and nature-based hardware. NP-GL lifts the binary limitation of existing nature-powered ML methods and extends their applicability to real-valued problems.
- We design a new hardware-friendly Hamiltonian for real-valued support, coupled with an efficient training method with two optimizations that ensure high training speed and quality.
- We develop a new nature-based computer for the new Hamiltonian, enabling nature's power in electronic dynamical systems to solve real-valued learning problems with extremely high speed.
- Experimental results across four real-world applications and six datasets show that NP-GL delivers $6.97 \times 10^3$ speedup and $10^5\times$ energy saving with even higher accuracy than GNNs.

## 2 BACKGROUND AND RELATED WORK

### 2.1 ISING MODEL AND ISING MACHINE

**The Ising model** is a probabilistic graphical model rooted in the statistical physics of ferromagnetism and widely employed to represent complex dynamical systems. Its Hamiltonian is as follows:

$$\mathcal{H}_{\text{Ising}} = -\sum_{i \neq j}^{N} J_{ij}\sigma_i\sigma_j - \sum_{i}^{N} h_i\sigma_i \; ; \; \sigma_i \in \{-1, +1\} \tag{1}$$

Due to magnetic physical property, an Ising spin $\sigma_i$ has two states "+1" and "−1", namely, spin up and spin down. $J_{ij}$ is the coupling parameter representing the spatial and temporal correlations among different spins; $h_i$ represents the self-reaction strength to the external impact and provides temporal continuity for time-series problems. The generalized Ising model goes beyond physical lattices and uses complete graphs to embed alll-to-all correlations among spins, therefore exhibiting strong connectivity, expressivity, and the long-range cascading ability to propagate information. A graph can be straightforwardly mapped to an Ising model, where the nodes are modeled as spins $\sigma$, and edges as a pair-wise coupling coefficient matrix $J_{ij}$ and the self-reaction $h_i$ of individual spins. In the realm of graph learning, these properties are exceptionally valuable for accuracy.

**Variations of the Ising model.** A few other models are related to the (vanilla) Ising model. The quantum Ising model allows spins to be in a state that is a superposition of up and down, incorporating quantum effects. It is used mostly for the study for quantum phase transitions (Kadowaki & Nishimori, 1998; Dziarmaga, 2005; Chakrabarti et al., 2008) and quantum magnetism (Labuhn et al., 2016; Schauß et al., 2015; Bitko et al., 1996). The XY model and the Heisenberg model extra degrees of freedom for spins so that they are unit vectors in the 2D and 3D spaces, respectively. The former is usually used to study topological phase transitions (Leoncini et al., 1998; Song & Zhang, 2021), and also acts as the model foundation for electronic oscillator-based Ising machines. The Heisenberg model allows spins to orient in three spatial dimensions, making it crucial for studying magnetism in certain materials (Fisher, 1964; Arnesen et al., 2001), while its 3D spin orientation significantly increases the model complexity. In spite of the number of Ising model variations, they largely remain theoretical with few practical realizations, as even realizing the vanilla Ising model is very challenging (more details introduced in (Afoakwa et al., 2021)). Reasonably, it is vital to preserve the feature of easy manufacturing when adding real-value support to the Ising model.

**Ising machine**, as a physical embodiment of the Ising model (Cipra, 1987), can be thought of as a dynamical system whose gradient is the Ising Hamiltonian. The inherent nature of gradient dynamical systems is to seek the lowest-energy or most probable states. Like other dynamical systems, the

Ising machine also naturally gravitates towards the lowest-energy state. What powers this inclination of the Ising machine to find the lowest-energy states is inherently derived from nature, more specifically, the spontaneous energy decrease in nature. To show some Ising machine examples, we list three prominent technologies, including D-Wave's quantum annealers (Harris et al., 2010), Coherent Ising Machines (CIMs) (Inagaki et al., 2016a), and coupled oscillators (Wang & Roychowdhury, 2019). D-Wave's system takes advantage of quantum effects with its superconducting qubits, but also faces constraints in problem mapping and high power consumption due to cryogenic operational requirements. CIMs use Optical Parametric Oscillators, enabling simpler, all-to-all spin coupling, but are challenged by scalability and temperature stability issues, as well as heavy computational demands for pulse modulation. In addition, emerging oscillator-based Ising machines, which form stable phase relationships and utilize LC tank oscillators suitable for analog circuits, face challenges in the oversized machine and the lack of high quality inductors. On the other hand, the Ising machine used as the backbone of the proposed NP-GL nature-based computer is BRIM (Afoakwa et al., 2021), the current SOTA. As discussed earlier, considering the difficulty in realizing the Ising models, even the vanilla model, the practicality of deployment becomes exceptionally important. Fortunately, BRIM is constructed to operate at room temperature using low-power circuits and in standard CMOS technology, inheriting decades of manufacturing experience. In such Ising machine, the states of objects, aka "spins", are effectively modeled as nano-scale capacitors and the coupling parameters representing the correlations among objects are modeled as resistors of varying resistance. Specifically, the electrons automatically traverse between capacitors through wires connected to resistors to establish a stable electric charge distribution, enabling the "speed of electrons". Consequently, the overall energy is minimized, and the lowest-energy state solution is found with extremely low operating latency. It is not hard to imagine that if the coupling parameters in the Ising Hamiltonian can be trained to effectively map the desired results of a given ML problem to the lowest-energy states, the Ising machine can then perform nanosecond-scale nature-powered ML inference and find the expected results.

## 2.2 ISING GRAPH LEARNING: BINARY GRAPH LEARNING BASED ON ISING MACHINE

When it comes to solving a graph learning problem using an Ising machine, two steps are encompassed. Firstly, the complete correlation graph is trained to construct an energy landscape, in which the energy ground state of the Ising model corresponds to the ground truth derived from historical data pertaining to the problem. Secondly, upon initializing the spins, the Ising machine undertakes a process of electron-speed annealing with asynchronous spin flipping, enabling it to efficiently converge towards the energy ground state that accurately represents the problem's ground truth. In essence, the computation is carried out by nature itself, resulting in an exceptionally rapid computational process. Recent studies have demonstrated that Ising Graph Learning outperforms GNN in binary problems. However, this new ML paradigm only works with binary problems.

## 2.3 GRAPH NEURAL NETWORKS

GNNs represent a class of models designed to address graph-structured data, which are prevalent across diverse domains, including physics(Shlomi et al., 2020), social science(Yang et al., 2021), bioinformatics(Yi et al., 2021), combinatorial optimization(LIU, 2022) and so on. GNNs effectively learn graph embeddings by iteratively propagating information between connected nodes in a graph. They usually employ a message-passing mechanism, where each node aggregates and updates information from its neighbors, allowing the network to capture complex relational patterns within the graph. These learned embeddings contain valuable insights about the graph, making GNNs a powerful tool (Xu et al., 2018; Zhou et al., 2020) for various graph-related tasks such as node classification(Kipf & Welling, 2016; Jiang & Luo, 2022), link prediction(Zhang & Chen, 2018; Chen et al., 2022), and graph classification(Wu et al., 2019a).

## 3 MOTIVATION: THEORETICAL ANALYSIS OF THE BINARY LIMITATION

This section theoretically examines the inherent binary limitation of the Ising model. Specifically, it demonstrates that the direct replacement of binary variables of the Ising Hamiltonian with real-valued ones without modifying the Hamiltonian formulation (namely, "naive real-valued Ising

model") will result in an energy landscape without local energy minima. Consequently, this model lacks the necessary expressivity to solve real-valued ML problems.

The stationary points of the naive real-valued Ising Hamiltonian are obtained by solving the below equation for $N$ spins, with $\sigma_i \in \mathbb{R}$ being the only difference from $\mathcal{H}_{\text{Ising}}$.

$$\frac{\partial \mathcal{H}_{\text{Ising}}^{\text{R}}}{\partial \sigma_i} = -\sum_{i \neq j}^{N}(J_{ij} + J_{ji})\sigma_j - h_i = 0 \tag{2}$$

which can be formulated into the following vector form:

$$-\mathbf{J}'\boldsymbol{\sigma} = \mathbf{h}; \ \ diag(\mathbf{J}') = \mathbf{0} \tag{3}$$

where $\mathbf{J}' = \mathbf{J} + \mathbf{J}^{\text{T}}$. For stationary point analysis, the Hessian Matrix of $\mathcal{H}_{\text{Ising}}^{\text{R}}$ is computed:

$$\mathbf{H}(\mathcal{H}_{\text{Ising}}^{\text{R}}) = -\mathbf{J}' \tag{4}$$

Because $\mathbf{J}'$ is a constant matrix, all stationary points have the same concavity. $\mathbf{J}'$ is also diagonalizable due to its symmetry. Therefore, based on the theorem in linear algebra, $tr(\mathbf{J}') = \sum_i^N \lambda_i$ where $\lambda_i$ is the $i$'th eigenvalue of $\mathbf{J}'$. As the diagonal line of $\mathbf{J}'$ is 0, both positive and negative eigenvalues exist unless $\mathbf{J}' = 0_{N \times N}$. This suggests that all the stationary points are saddle points instead of local minima, indicating that there are always neighbor states with lower energy and therefore no local minima can be found by the Ising machine. In practice, as both the energy of the Ising model and the values of the spins in the Ising model have boundaries (e.g., +1 and -1 for the spins), the Ising machine always stops at the boundary energy state with highly polarized spin values. If used in ML, these spin values are uninformative, normally representing inaccurate results. To this end, we are convinced that the Hamiltonian must be upgraded to support real values.

In addition to the Ising model, adjustments in the Ising machines are also desired. Constructed upon the Ising model, Ising machines specialize in accurately bisecting a group of nodes into two parties with opposite features, putting the effort in maintaining a polarized result. Nevertheless, for the sake of real-valued support, it is favorable to allow the spins to stabilize at intermediate states rather than being exclusively polarized at their boundaries. By implementing this adjustment, the machine is granted the ability to express a wider range of values and is therefore generally applicable in ML.

To bring the nature power to the realm of real values, we are motivated to break the binary limitations by developing the NP-GL framework, which is described in depth in the following section.

## 4 THE PROPOSED NP-GL FRAMEWORK

This section introduces the details of the proposed NP-GL framework. We first provide an overview of NP-GL workflow, and then step-by-step present the design of NP-GL Hamiltonian, training algorithm, and nature-based hardware architecture.

### 4.1 OVERVIEW OF NP-GL

Figure 2 shows three essential steps in solving real-world graph learning problems with NP-GL. First, a real-world graph is mapped to our Hamiltonian-based probabilistic graphical model, which is upgraded from the Ising Hamiltonian to support real values (Sec. 4.2). In the model, the nodes are modeled as spins, while the relations between spins, or logical edges, are modeled as coupling and self-reaction parameters $\mathbf{J}$ and $\mathbf{h}$. Second, through training (Sec. 4.3) with real-valued historical data, an energy landscape is constructed by learning the Hamiltonian parameters, during which the energy ground state is mapped to the maximum likelihood in the model determined by the observed samples. Third, to align with the upgrade of Hamiltonian, we make slight modifications to a SOTA Ising machine to build a nature-based computer for NP-GL. After the trained parameters are deployed on our nature-based computer, the computer starts to search for the lowest energy state that represents the desired inference results (Sec. 4.4). The specific computing process is as follows: For the known spin values, we keep the spins fixed to the known real-valued data and let other spins evolve to find the lowest energy state conditioned on the known information. For temporal graph prediction, spins are uniformly divided into multiple partitions, representing the nodes from the previous, current, and future time steps, respectively. The nodes representing the previous and current time steps are fixed to historical data, while the others are predicted by evolving towards the spin configuration with the lowest energy.

Figure 2: The workflow of NP-GL, involving Hamiltonian mapping, training, and inference steps.

## 4.2 THE PROPOSED HAMILTONIAN

The new Hamiltonian is proposed for three major reasons. (1) To solve the spin polarization problem in Sec. 3. (2) To enable effective training techniques. (3) To minimize the adjustment of our backbone hardware. We hereby use a pure quadratic term to replace the linear term in Ising Hamiltonian:

$$\mathcal{H}_{\text{RV}} = -\sum_{i \neq j}^{N} J_{ij} \sigma_i \sigma_j + \sum_{i}^{N} h_i \sigma_i^2, \ \sigma_i \in \mathbb{R} \tag{5}$$

It can be inferred that $\mathbf{J}$ and $\mathbf{h}$ still represent spin coupling and self-reaction similar to the Ising model, inheriting the connectivity, expressivity, and the information propagation ability from it. Additionally, this Hamiltonian can be formulated more compactly as $\mathcal{H}_{\text{RV}} = -\sum_{i,j}^{N} J_{ij} \sigma_i \sigma_j$ with $J_{ii} = -h_i$, indicating that $(-\mathbf{h})$ is embedded in $\mathbf{J}'$ as the diagonal line. If $\mathbf{h}$ is positive, the convexity of the Hamiltonian is ensured, enabling the existence of local minima. As we will discuss shortly, $\mathbf{h}$ is guaranteed positive with our training method, and the adjustment in Hamiltonian can be easily implemented on top of our backbone hardware.

## 4.3 THE TRAINING METHOD OF NP-GL HAMILTONIAN

The training for Hamiltonian parameters aims to map the energy ground state to the maximum likelihood with real-valued data. It consists of two major components: the conditional likelihood method as the backbone, and two optimization methods to enhance training quality.

### 4.3.1 THE CONDITIONAL LIKELIHOOD METHOD AS BACKBONE

To perform real-valued training with affordable computing power, a conditional likelihood method is proposed. The spin configuration of our model is denoted as $S = \{\sigma_1, \ldots, \sigma_N\}$. The joint probability density function of spin configuration $S$ is defined under the Boltzmann distribution

$$p(S|\mathbf{J}, \ \mathbf{h}) = Z^{-1} \exp\{-\mathcal{H}_{\text{RV}}\} \tag{6}$$

where $Z$ is the partition function. Instead of maximizing the likelihood for the observed samples altogether which is computationally infeasible, a conditional likelihood is maximized as an approximation. That is to say, we optimize the parameters by focusing on one spin (say, $\sigma_i$) at a time, while treating other spin values as conditions. In particular, we split $\mathcal{H}_{\text{RV}}$ into $\mathcal{H}_{\text{RV},i}$ and $\mathcal{H}_{\text{RV},\backslash i}$ to represent the terms dependent and independent of $\sigma_i$. The joint probability density then becomes:

$$p(\sigma_i, \ \sigma_{\backslash i}) = Z^{-1} \exp\{-\mathcal{H}_{\text{RV},i} - \mathcal{H}_{\text{RV},\backslash i}\} \tag{7}$$

If consider all $\sigma_{\backslash i}$ as conditions, $\mathcal{H}_{\text{RV},\backslash i}$ becomes a constant. Subsequently, we obtain:

$$p(\sigma_i = \mu | \sigma_{\backslash i}) \propto \exp\{-\mathcal{H}_{\text{RV},i}|_{\sigma_i = \mu}\} \tag{8}$$

Specifically,

$$\mathcal{H}_{\text{RV},i}|_{\sigma_i = \mu} = -\varepsilon_i \mu + h_i \mu^2 \tag{9}$$

where $\mu$ is the spin variable, and $\varepsilon_i = -\sum_{j \neq i}^{N} J_{ij} \sigma_j$ for conciseness. It is apparent that $\mathcal{H}_{\text{RV},i}$ is minimized at $\mu = \varepsilon_i / 2h_i$, which is determined by the trained parameters and the other conditional spin values. This indicates that the value of a spin can converge between its upper and lower bounds, fundamentally supporting real-valued data. To train this model, either the Mean Square Error (MSE) loss or the L1 loss between $\mu = \varepsilon_i / 2h_i$ and the ground truth is utilized to maximize the conditional likelihood. Once the training is complete, the maximum conditional likelihood is obtained, which is automatically mapped to the energy ground state according to equation 8

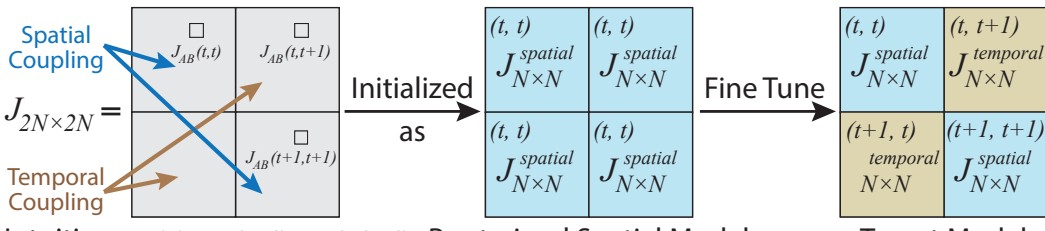

Figure 3: Fine-tuning of a pre-trained spatial model. $J_{AB}$: the coupling between Spin $A$ and $B$.

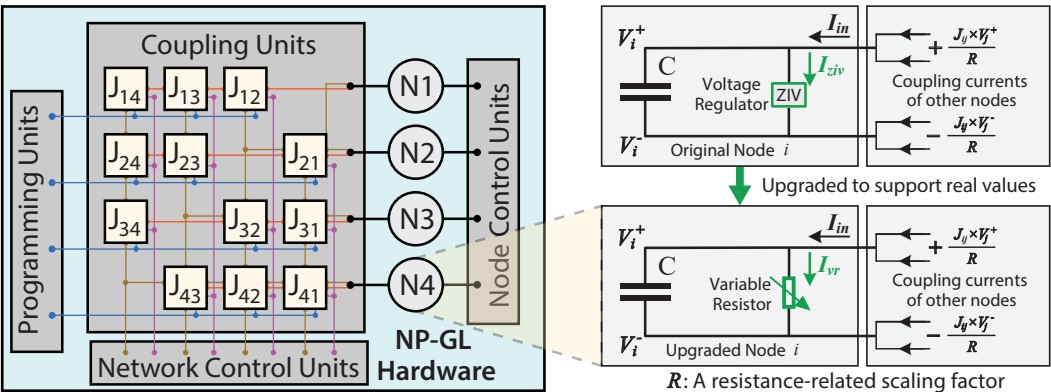

Figure 4: The nature-based computer upgraded from its binary predecessor to support real values.

This training approach is applicable to both spatial and spatial-temporal models. In the former scenario, the spins only represent the nodes at the same time step, while in the latter case, more spins are necessary to represent the nodes from past, current, and future time steps.

### 4.3.2 TRAINING ENHANCEMENT TECHNOLOGIES

With the training backbone established, there are still two questions remain to be answered. (1) **J** and **h** have distinct physical meanings. Is there a way to differentiate them during training? (2) How do we leverage the similarity between spatial and temporal correlations?

**A. Diagonal Line Reinforcement (DLR):** **J** and **h** are trained jointly with the training method described above. However, these two types of parameters have distinct physical meanings: coupling and self-reaction. In practice, they also usually differ by orders of magnitude. To differentiate these parameters in training, we reinforce **h** on the diagonal line of **J** by multiplying a scaling factor as a hyperparameter. This method is applicable to both spatial and spatial-temporal models. As Figure 1 illustrates, the diagonal line of $\mathbf{J}_{spatial}$ is directly multiplied by the factor to reinforce the self-reaction. Similarly, in a spatial-temporal model, all four submatrices have their individual diagonal lines reinforced, enhancing both the self-reaction and the temporal coupling of the same node.

**B. Spatial Pre-training Coupled with Temporal Fine-tuning:** Intuitively, Spin $A$ at time $t$ and $t+1$ may have similar properties. For instance, they may have similar coupling relations with respect to Spin $B$, as Figure 3 shows. Based on this, we speculate that the temporal couplings may have similar values as the spatial couplings. To exploit this property, we pre-train a spatial model featured as $J_{N \times N}$ and map it onto a spatial-temporal model as initial values. The model is subsequently fine-tuned to obtain a final model. Through this approach, the similarity between spatial and temporal correlations is leveraged, resulting in improved accuracy and convergence speed.

### 4.4 INFERENCE WITH A NEW NATURE-BASED COMPUTER

Aiming to harness the power of nature and achieve an end-to-end solution for graph learning, the inference process is carried out on a nature-based computer modified from a SOTA Ising machine to align with the upgrade in the Hamiltonian. The overall layout of the computer is shown in Figure 4, in which the nodes are connected through the coupling units in an all-to-all manner. Specifically,

Table 1: Accuracy comparison (in MAE / RMSE): the lower the better - best results are in bold.

| Application | Traffic Flow | | Air Quality | | Taxi Demand | Pandemic Progression |
|---|---|---|---|---|---|---|
| Dataset | PEMS04 | PEMS08 | PM2.5 | PM10 | NYC Taxi | Texas COVID |
| Graph WaveNet | 20.84 / 33.66 | 15.77 / 24.03 | 1.823 / 3.106 | 1.954 / 3.496 | 10.22 / 21.24 | 82.96 / 430.1 |
| ASTGCN | 20.79 / 33.39 | 15.68 / 23.73 | 1.883 / 3.133 | 2.255 / 3.662 | N/A[2] | N/A |
| MTGNN | 19.96 / 31.64 | 15.15 / 22.79 | 1.833 / 2.933 | 1.990 / 3.303 | 7.079 / 15.40 | 84.17 / 414.2 |
| DDGCRN | 18.97 / 30.59 | 14.64 / 22.42 | 1.711 / 3.004 | 1.881 / 3.315 | 3.059 / 10.21 | 23.94 / 188.8 |
| MegaCRN | 17.65 / 29.25 | 13.70 / **21.03** | 1.646 / 2.863 | 1.741 / 3.098 | 6.082 / 15.01 | 83.73 / 423.6 |
| NP-GL-Vanilla-MSE | 18.37 / 29.10 | 16.02 / 24.31 | 1.710 / 2.868 | 1.860 / 3.146 | 3.741 / 11.41 | 41.46 / 268.9 |
| NP-GL-MSE | 18.09 / 28.66 | 15.11 / 23.40 | 1.702 / **2.859** | 1.862 / 3.144 | 3.195 / **9.610** | 37.60 / 268.8 |
| NP-GL-L1 | **17.07 / 27.66** | **13.51** / 21.69 | **1.624** / 2.914 | **1.730 / 3.013** | **3.031** / 10.08 | **22.04 / 117.7** |

the spin values are represented as the voltage on the capacitors in the nodes, while the parameters $\mathbf{J}'$ and $\mathbf{h}$ are represented as conductance. On the right-hand side of the figure, the modifications are colored green, showing that we merely replace the voltage regulator "ZIV" with a variable resistor in each node to import the quadratic term $\sum_i h_i V_i^2$ into our Hamiltonian. The electric current of Node $i$ is written as equation 10, where $J_{ij}'$ is the effective conductance of the coupling between nodes and $2h_i$ is the effective conductance of the added variable resistor in the node.

$$\frac{dV_i}{dt} = \frac{1}{C}(I_{in} - I_{vr}) = \frac{1}{C}(\sum_{j \neq i} J_{ij}'V_j - 2h_iV_i) = -\frac{1}{C}\frac{\partial \mathcal{H}_{\mathrm{RV}}}{\partial V_i} \quad (10)$$

which satisfies

$$\frac{d\mathcal{H}_{\mathrm{RV}}}{dt} = \sum_i (\frac{\partial \mathcal{H}_{\mathrm{RV}}}{\partial V_i}\frac{dV_i}{dt}) = -C\sum_i (\frac{dV_i}{dt})^2 \leq 0 \quad (11)$$

This suggests the Hamiltonian described by the CMOS compatible circuit spontaneously decreases. In other words, the lowest energy state is automatically pursued.

After the Hamiltonian parameters $\mathbf{J}$ and $\mathbf{h}$ are obtained from training, they are deployed on our nature-based computer for inference. In practice, the spins representing the nodes from the previous and current time steps are fixed to the known data, allowing the spins representing the nodes in future to evolve and reach the spin configuration with the lowest energy, thus performing the prediction.

# 5 EXPERIMENTAL RESULTS

## 5.1 EXPERIMENTAL SETUP

**Applications and Datasets.** We evaluate the NP-GL framework with four real-world spatial-temporal applications and six real-world datasets including Traffic Flow, Air Quality, Taxi Demand, and Pandemic Progression as follows. More information is reported in Appendix A.1.

- Application: *Traffic Flow* – Prediction of the number of vehicles passing through detectors per unit time; Datasets: *PEMS04 & PEM08* – Traffic flow data in metropolitan areas of California.
- Application: *Air Quality* – Prediction of air pollution levels; Datasets: *CAQRA-PM2.5 & CAQRA-PM10* – Chinese historical PM2.5 and PM10 data from 2019.5 to 2019.12 sampled from the Chinese Air Quality Reanalysis (CAQRA) database.
- Application: *Taxi Demand* – Prediction of the hourly number of taxi trips; Dataset: *NYC Taxi* – The hourly number of taxi trips in New York City in 2022.
- Application: *Pandemic Progression* – Prediction of the daily number of new cases; Dataset: *Texas COVID* – 2020-2023 daily case increment of COVID-19 in Texas.

**Baselines.** For fair evaluation, we compare NP-GL accuracy and inference latency with five SOTA spatial-temporal GNNs including *Graph WaveNet* (Wu et al., 2019b), *ASTGCN* (Guo et al., 2019), *MTGNN* (Wu et al., 2020), *DDGCRN* Weng et al. (2023), and *MegaCRN* Jiang et al. (2023). **Platforms.** NVIDIA A100-40GB GPUs and AMD Ryzen Threadripper PRO 3975WX CPUs are used to measure the inference latency of the baseline GNNs. The nature-based computer utilized to evaluate the performance of NP-GL is adapted from the current SOTA Ising machine, BRIM (Afoakwa et al., 2021), based on Cadence Analog Design Environment.

---

[2]Graph topology is unavailable for NYC Taxi and Texas COVID datasets. Different from MTGNN and Graph WaveNet, ASTGCN does not support graph topology generation.

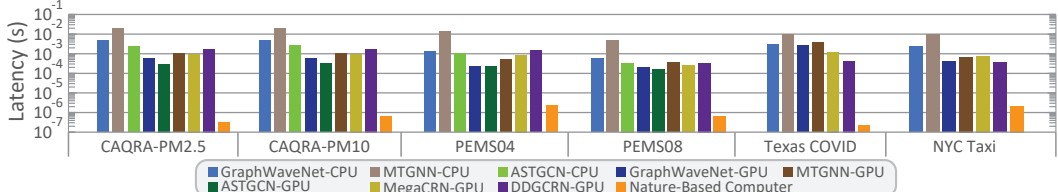

Figure 5: The latency comparison of CPU, GPU, and NP-GL nature-based computer.

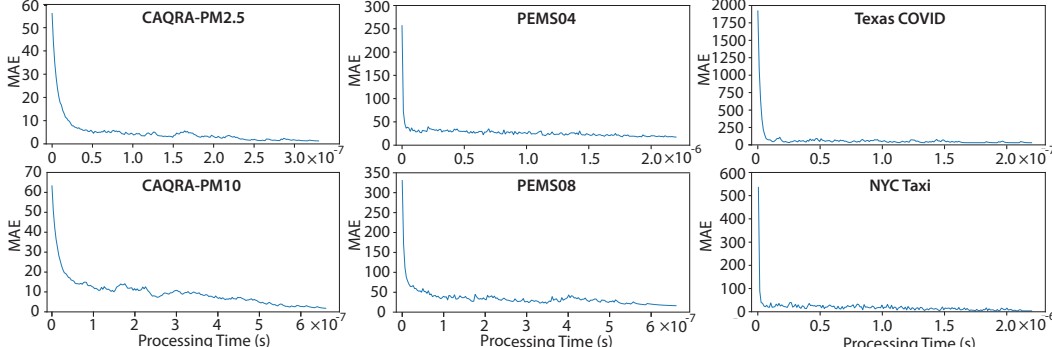

Figure 6: Natural process of energy decrease in NP-GL computer to find desired inference results.

## 5.2 EVALUATION OF ACCURACY

Table 5.2 compares the accuracy of predicting the $t$'th snapshot based on the $(t-1)$'th snapshot with five GNN baselines and two NP-GL variants. NP-GL-Vanilla-MSE is equipped with the DLR optimization. NP-GL-MSE is further improved with Spatial Pre-training and Temporal Fine-tuning. NP-GL-L1 uses L1 loss instead of MSE loss compared with NP-GL-MAE, as we realize that the data outliers may significantly affect the accuracy. Mean Absolute Error (MAE) and Root Mean Squared Error (RMSE) that reflect the average and the deviation of the result quality are adopted as the accuracy metrics for comparison. For the MAE metric, NP-GL outperforms all 5 baselines across all datasets. For RMSE, except for PEMS08 where MegaCRN is slightly better, NP-GL outperforms all other baselines.

## 5.3 EVALUATION OF LATENCY

This section evaluates the inference latency of NP-GL. Figure 5 compares the inference latency of GNNs and NP-GL, highlighting the extraordinarily short latency introduced by our nature-based computer that is powered by nature and operates at extremely high speed. In the figure, the nature-based computer delivers orders of magnitude speedup (from $10^3$ to $10^4$) compared to CPU and GPU results across all datasets and baselines. Specifically, $1.70 \times 10^4$ speedup is achieved on average for the Texas COVID dataset, while the dataset with the least speedup (NYC Taxi) still impressively reaches $1.01 \times 10^3$. To highlight a comprehensive result, we take the mean value of all 6 average speedups to obtain the overall average speedup as $6.97 \times 10^3$. Figure 6 shows spin evolution processes of the nature-based computer throughout inference. As the spins evolve towards the lowest energy state, the error of the results found by the computer decreases accordingly. The curves show the nature-based computer rapidly optimizes the spin configurations to approach the lowest energy states which are also the desired learning solutions. In terms of energy consumption, our nature-based computer operates at a power of $\sim 500 \ mW$, more than $100\times$ lower than modern GPUs and CPUs. The overall energy consumption is approximately $10^5$ less than GPUs and CPUs.

## 6 CONCLUSION AND FUTURE INSIGHTS

This paper extends the computational power of nature to real-world graph learning problems by proposing NP-GL. Specifically, NP-GL is a nature-powered graph learning model and solves real-valued graph learning problems by harnessing the natural phenomenon of energy decrease. Experimental results across 4 real-world applications with 6 datasets demonstrate that NP-GL delivers, on average, $6.97 \times 10^3$ speedup and $10^5$ energy consumption reduction with comparable or even higher accuracy than GNNs.

ACKNOWLEDGEMENTS

This work was supported by NSF under Awards SHF-2326494 and No.2231036; by NYS Center of Excellence under Awards No.2089C001; and by META Reality Lab. This research was also partially supported by the U.S. DOE Office of Science, Office of Advanced Scientific Computing Research, under award No.66150: "CENATE - Center for Advanced Architecture Evaluation" and No.78284: "ComPort: Rigorous Testing Methods to Safeguard Software Porting".

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

# A APPENDIX

## A.1 DATASETS

NP-GL is evaluated with four real-world spatial-temporal applications and six real-world datasets listed below. Their specification details is reported in Table A.1. All datasets are partitioned into 70/20/10 percentages for training, validation, and test purposes.

Table A.1: Dataset specifications. A snapshot is the state of the data at a specific point in time.

| Application | Traffic Flow | | Air Quality | | Taxi Demand | Pandemic Progression |
|---|---|---|---|---|---|---|
| Dataset | PEMS04 | PEMS08 | CAQRA-PM2.5 | CAQRA-PM10 | NYC Taxi | Texas COVID |
| # of Snapshots | 16992 | 17856 | 5856 | 5856 | 8759 | 1100 |
| # of Nodes | 307 | 170 | 400 | 400 | 259 | 256 |
| Data Range (Max-Min) | 919 | 1147 | 167.0 | 186.9 | 1115 | 25190 |

## A.2 HYPERPARAMETERS

For all experiments carried out on the six datasets (detailed specifications reported in Table A.1), the Rprop optimizer is adopted with a custom early-stop mechanism. The same hyperparameters are used across the experiments except for the DLR factors used to reinforce diagonal lines of $\mathbf{J}$, which are listed in Table A.2.

Table A.2: DLR Factors as the Hyperparameters for Different Datasets.

| Application | Traffic Flow | | Air Quality | | Taxi Demand | Pandemic Progression |
|---|---|---|---|---|---|---|
| Dataset | PEMS04 | PEMS08 | PM2.5 | PM10 | NYC Taxi | Texas COVID |
| DLR Factor | 12.28 | 8.50 | 16.00 | 6.00 | 12.95 | 5.89 |

Table A.3: The universal hyperparameters used for all 6 datasets.

| Minibatch | LR | Etas | Step Sizes | LR Decay | Step Decay | Early Stop |
|---|---|---|---|---|---|---|
| 64 | 0.01 | (0.5, 1.2) | (1E-4, 50) | 0.5 | 50 | 1E-6 |

The rest of the hyperparameters are listed in Table A.3, where LR is the initial learning rate, Etas and Step Sizes are the hyperparameters intrinsic to the Rprop optimizer. To implement an early stop mechanism, the learning rate is multiplied by 0.5 (LR Decay) if the loss function does not decrease in 50 epochs (Step Decay). Once the learning rate is below 1E-6 (Early Stop), the training process is terminated.

## A.3 DATA DISTRIBUTION PROFILING

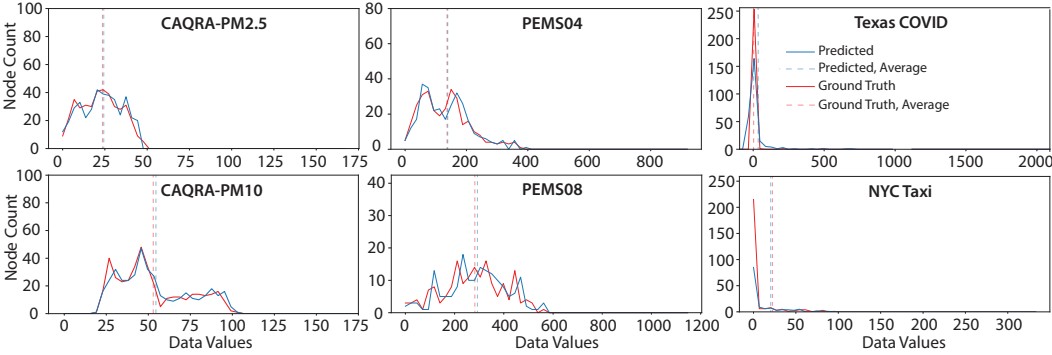

Figure A.1: The data distribution comparison of the predicted data and ground truth data.

The example predicted data distribution and the ground truth distribution across all 6 datasets are illustrated in Figure A.1. The examples are randomly sampled from the snapshots used for graph prediction. It can be observed that despite the datasets have distinct data ranges and distributions, the predicted data distribution obtained from our approach is very close to the ground truth, with the mean data values (dashed lines) almost identical.

