# OpenReview forum: "Extending Power of Nature from Binary to Real-Valued Graph Learning in Real World"
_ICLR.cc/2024/Conference — ICLR 2024 poster_

### Official Review · Reviewer_qast · 2023-10-31

**Soundness:** 3 good
**Presentation:** 3 good
**Contribution:** 3 good
**Rating:** 6
**Confidence:** 4

**Summary:**

This paper tackles the issue of the binary nature of existing Ising graph learning by proposing a new Hamiltonian for real-value graph learning problems.

**Strengths:**

* The paper is well-written and easy to follow, more like a report.
* Solve an important problem in current Ising graph learning using nature-law-based machines, i.e., only supporting binary values.
*  Efficient training methods with three optimizations that ensure training speed and quality.
* Achieves the best accuracy compared to three baselines

**Weaknesses:**

* Over-exaggerated descriptions in the main context. The paper seems to overclaim some parts, like mentioning, “Regrettably, despite their perceived potential, most
nature-powered ML methods are still predominantly theoretical, outperforming NNs only in toy problems under highly idealized conditions” while the Ising machine is also not quite partial and only applicable for some problems.
* Out-dated baselines. In comparison, the authors chose three “SOTA” spatial-temporal GNNs, while the earliest was published in 2020. The author should compare with more recent advances.

**Questions:**

*  Could the authors provide more comparison with recent SOTA GNN spatial-temporal GNNs? I found one recent paper,
    * Jiang, Renhe, et al. "Spatio-temporal meta-graph learning for traffic forecasting." Proceedings of the AAAI Conference on Artificial Intelligence. Vol. 37. No. 7. 2023.
* What’s the physical meaning of asymmetric weight decomposition? As it is a natural process (Hamitonian), is this symmetric weight decomposition meaningful for the physical system besides the efficiency consideration?

---

> ### Author Response · Authors · 2023-11-19
>
> We sincerely thank the reviewer for their valuable comments and questions. We also provide a "general comment" for all reviewers at the top as a global rebuttal other than the specific answers below.
>
> **Comment 1**. Over-exaggerated descriptions.\
> **Response:**
>
> We sincerely thank the reviewer for this comment. We have revised the sentence pointed out by the reviewer and will comprehensively revise the paper and adjust the tone. Again, we thank the reviewer for pointing out this.
>
> **Comment 2 & Question 1**. More recent baseline.\
> **Response:**
>
> The reviewer's suggestion is very much appreciated. We have added 2 more recent work, including the work the reviewer suggests in our revision. See table 2. To enrich the evaluation, we also added more latency results and accuracy results of NP-GL equipped with different loss functions for better comparison. The corresponding modifications are highlighted with label "Reviewer-qast"
>
> **Question 2**. Physical meaning of symmetric weight decomposition.\
> **Response:**
>
> The symmetric weight decomposition acts as a natural constraint such that the spin correlation matrix (the "J'" matrix in context) becomes symmetric. In analogy to Newton's 3rd Law, the forces applied to the objects in a pair are the same in magnitude. In the real-world, most of dynamic systems have pair-wise connection among nodes. Such symmetry is very common in physics, including Ising model. To explain in the language of math, after taking the first derivative of the Hamiltonian, the symmetric term (J+J transpose) appears, indicating the symmetric nature of the correlation matrix.

---

> > ### Comment · Reviewer_qast · 2023-11-22
> >
> > Thanks for the response. A new series of main results, which can beat the recent baselines, are listed in the rebuttal phase. I will raise my point, but please revise your description carefully and adjust the tone.

---

> > > ### Author Response · Authors · 2023-11-22
> > >
> > > Thank you so much for your time and encouragement! Also much appreciated for the insightful comments and suggestions to help improve the quality of this paper. We will very carefully work on the descriptions and lower the tone.

---

> ### Author Response · Authors · 2023-11-22
>
> Thank you again for the constructive suggestion! We have revised the description and lowered the tone in the newly uploaded revision for your review (shaded in orange). Your time is very appreciated, and any further comments are very much welcomed. Also, we thank you in advance for your continued efforts in the rest of the review process. If you are celebrating it, we wish you a wonderful Thanksgiving!

---

### Official Review · Reviewer_w2U6 · 2023-11-01

**Soundness:** 3 good
**Presentation:** 3 good
**Contribution:** 3 good
**Rating:** 6
**Confidence:** 3

**Summary:**

**Objective:** The paper aims to extend the capabilities of nature-powered computations from binary problems to real-valued graph learning problems. The authors propose a novel end-to-end Nature-Powered Graph Learning (NP-GL) framework, which leverages the natural power of entropy increase to efficiently solve real-valued graph learning problems.

**Methodology:** The NP-GL framework is designed through a three-dimensional co-design, incorporating a new Hamiltonian that is hardware-friendly, maintains distinct stable states with real values, and ensures high expressivity. The training algorithms of NP-GL adopt an improved conditional likelihood method with optimizations for complexity reduction, convergence expediation, and better learning from temporal information. Additionally, a new nature-based computer is developed to support the NP-GL Hamiltonian, enabling the solution of real-valued graph learning problems.

**Results:** Experimental results across four real-world applications and six datasets demonstrate that NP-GL delivers, on average, a 6.97 × 10^3 speedup and 10^5× energy consumption reduction, with comparable or even higher accuracy than Graph Neural Networks (GNNs).

The contribution lies in

**Extending Nature-Powered ML to Real-Valued Problems:** The paper introduces NP-GL, an end-to-end nature-powered graph learning method that breaks the binary limitation of existing nature-powered ML methods, extending their applicability to real-valued problems.

**New Hardware-Friendly Hamiltonian:** A new Hamiltonian is designed for real-valued support, coupled with an efficient training method that ensures high training speed and quality.

**Development of a New Nature-Based Computer:** A new nature-based computer is developed for the NP-GL Hamiltonian, using the Ising machine as a backbone, enabling the solution of real-valued graph learning problems using nature’s power.

**Significant Speedup and Energy Savings:** NP-GL demonstrates a substantial speedup (6.97 × 10^3) and energy savings (10^5×) compared to GNNs, with even higher accuracy across various real-world applications and datasets.

**Strengths:**

1. Originality:
- Innovative Approach: The paper introduces a novel end-to-end Nature-Powered Graph Learning (NP-GL) framework, extending the capabilities of nature-powered computations from binary problems to real-valued graph learning problems. This is a significant departure from existing nature-powered ML methods, showcasing a high level of originality.
- Unique Integration: The three-dimensional co-design integrating a new Hamiltonian, training algorithms, and a nature-based computer is a unique approach that has not been explored extensively in previous works.
2. Quality:
- Robust Methodology: The paper employs a robust methodology, incorporating a hardware-friendly Hamiltonian, efficient training methods with optimizations, and the development of a new nature-based computer.
- Comprehensive Evaluation: The experimental results across four real-world applications and six datasets provide a comprehensive evaluation of the NP-GL framework, demonstrating its effectiveness in delivering significant speedup, energy savings, and high accuracy compared to GNNs.
3. Clarity:
- Well-Structured: The paper is well-structured, with a clear introduction, background, methodology, results, and conclusion sections. This structure aids in the reader’s understanding of the content.
- Detailed Explanations: The authors provide detailed explanations of the NP-GL framework, the new Hamiltonian, the training algorithms, and the nature-based computer, ensuring that readers can grasp the complexities of the work.
4. Significance:
- Addressing Real-World Problems: By extending the applicability of nature-powered ML methods to real-valued problems, the paper addresses a significant gap in the field, making it highly relevant to real-world applications.
- Potential for Impact: The demonstrated speedup, energy savings, and accuracy improvements have the potential to make a substantial impact in the field of graph learning, showcasing the significance of the work.

**Weaknesses:**

- Insufficient Discussion on Challenges: While the paper provides a comprehensive overview of the NP-GL framework and its benefits, there could be a more in-depth discussion on the potential challenges and limitations of the proposed approach. Providing such insights would offer a balanced view and help guide future research.

Minor: Add one-liner for the future insights summary to main text from Appendix.

**Questions:**

- Could you elaborate on the adaptation for NP-GL on top of SOTA Ising machine? The hardware implementation, challenges, and potential optimizations could provide valuable insights for readers interested in the practical aspects of the work.
- To confirm I assume the the engergy and latency are based on the simulation in CAD software?
- Would this machine be able to generalize beyond GNN?

- How it performs as the size of the graph increases and any potential strategies for handling large-scale graph learning problems?

---

> ### Author Response · Authors · 2023-11-19
>
> We sincerely thank the reviewer for their insightful questions. Apart from the specific answers to the questions listed below, we also include a "general comment" attached at the top as a global rebuttal.
>
> **Comment 1 & Question 1**. Limitations and more insights of implementation, challenges, and potential optimizations.\
> **Response:**
>
> 1. **Area:** Although it is much smaller than many other types of nature-based computer, e.g. oscillator-based Ising machines, the major limitation of the proposed method is still the area of machine. The chip is around 5mm^2, much smaller than GPUs and CPUs. However, as a chip with less than 1W power, the area is large. The reason is that the chip is predominantly based on analog circuits that are naturally larger than digital chips with the same power. If the chip works alone, the area is not an issue. However, if the proposed chip is to be integrated with other computers, the area problem is not negligible. For example, integrating the proposed nature-based hardware within GPUs (like tensorcores as a co-processor of CUDA cores) will equip GPUs with huge extra computational capability, but the chip area can be too big to fit in for now.
>
> 2. **Formulation:** Currently, due to the complexity of the coupling network, the NP-GL hardware is only capable of representing linear and quadratic terms within the Hamiltonian function. Considering the practicality in chip manufacturing, the quadratic limitation is untapped now. This may be a limitation on accuracy, although quadratic terms perform well with most of the applications we investigated (probably due to dynamic systems in the real-world being mostly pairwise). We are studying how to realistically add complexity to the current Hamiltonian.
>
> We have added this discussion of weakness in the Appendix of the paper due to page limit. See highlight labeled as "Reviwer-w2U6" in the "Conclusion and Future Insights" section and Appendix A.5.
>
> **Question 2**. Are the energy and latency based on CAD software?\
> **Response:**
>
> Yes. We will try to provide results based on the actual chip (being taped out) in the final version.
>
> **Question 3**. Would this machine be able to generalize beyond GNN?\
> **Response:**
>
> Yes. Here we list a few examples to demonstrate the deeper influences of our framework. Actually, NP-GL can generate unknown data based on a small amount of observed data. It reads training samples to understand the data distribution of a certain problem and generates data based on probabilities of the distribution. For graph prediction as an example, the observed data are historical time frames, and the unknown data are to be predicted. As NP-GL can impute unknown data as described, it can reasonably do more. Followed are some examples that we have largely validated, though not the focus of the paper.
>
> 1. **Data compression.** This framework can complete data based on the partial data stored in memory. This will save bandwidth and storage space, leading to higher performance. For example, this framework can impute a gradient map of deep learning training with 20% known gradient data points rather accurately. Therefore, with this framework, the storage and bandwidth requirement of deep learning training will be reduced by 5x.
>
> 2. **3D synthesis.** We already have some preliminary results showing this framework can be used for 3D synthesis and, although so far with lower quality than NeRF, construct pictures with micro-second latency. This is especially useful in AR/VR development, as real-time rendering is highly demanded. Here, the polar and azimuthal angles and some seed pixels are used as the observed data, with the remaining unknown pixels to be completed based on the observed data.
>
> Even within the scope of GNN, this framework can be used to improve GNN. For example, given a graph, we can use GNN to perform inference for only 10% nodes, while letting NP-GL complete the remaining 90% nodes based on the prior 10% results. In addition to speedup and reduction in storage and communication, if the 10% nodes are carefully chosen, e.g., the nodes of the highest accuracy in GNN, the overall accuracy can be improved.
>
> **Question 4**. How to handle larger graphs?\
> **Response:**
>
> If the graph is too large, we generally have two solutions: (1) temporal co-annealing – different nodes are mapped onto the chip for annealing at different time, such that the originally continuous annealing process is broken into several iterative steps. In this scenario, the strongly connected nodes should anneal at the same time for better convergence. (2) multi-chip integration - the system can be scaled up with multiple chips connected, with the strongly connected nodes mapped onto the same chip. For both solutions, there may be slowdowns. Nevertheless, for real-world graphs with community structure, the slowdown is acceptable if the nodes in the same community can be annealed simultaneously or on the same chip (less communication for synchronization).

---

> > ### Author Response · Authors · 2023-11-22
> >
> > As the end of the discussion session approaches, we would like to express our sincere gratitude for all the constructive and insightful reviews and suggestions. Any further comments on our responses and revisions are very much welcomed! We also thank you in advance for your continued efforts in the rest of the review process. If you are celebrating it, we wish you a happy Thanksgiving!

---

### Official Review · Reviewer_UYh4 · 2023-11-01

**Soundness:** 3 good
**Presentation:** 2 fair
**Contribution:** 3 good
**Rating:** 5
**Confidence:** 4

**Summary:**

This paper propose a novel graphic model architecture by improving the existing ising model. The authors further propose the efficient training and inference algorithm to search for the local minimum of the proposed solution. The paper also shows the potential hardware architecture to implement this novel graphical model. The results demonstrate that the proposed solution can achieve comparable performance as GNN over multiple tasks.

**Strengths:**

+ A novel graphical model architecture
+ Inference and training methods to achieve the local min
+ Efficient hardware implementation

**Weaknesses:**

- To me, the most significant problem of this work is insufficient review for the prior work. Section 2.1 and 2.2 is a good background introduction, but not too many prior works on the variation of Ising model are discussed, making the contribution of this work hard to justify.
- A section is missing to introduce the prior work on hardware implementation for Ising model.
- Technically, this paper simply propose a modified version of Ising model by using a pure quadratic term to replace the linear term in Ising Hamiltonian.
- Graphic model may not be well-suited for ICLR, which mostly focus on deep learning.

**Questions:**

Please see the weakness section and solve the problem accordingly.

---

> ### Author Response · Authors · 2023-11-19
>
> We sincerely thank the reviewer for their valuable insights and comments. Our answers to the specific questions are listed below, and a "general comment" is also provided for all reviewers to better express our ideas (attached at the top as a global rebuttal)
>
> **Comment 1 & 2**. More discussion of prior work.\
> **Response:**
>
> Thank you sincerely for pointing this weakness out. We have drafted a new related work section in the paper (section 2). Please see the highlighted text with label "Reviewer-UYh4". In addition, we provide more information below about the development stage of using Ising model and Ising machine in solving real-world problems.
>
> We believe this is a good time now to study novel learning methods based on the Ising model due to the recent emergence of efficient Ising machines. Although the Ising model has been famous and studied for decades, especially in statistical physics (widely used to describe complex dynamic systems), its related research largely remains on the theory level and rarely can be adopted in practice. One of the major reasons is the extreme complexity in solving an Ising model (find the lowest-energy states of a dynamic system) with a large variable space. From the perspective of computational time, solving the Ising model with a digital processor can be over 1000x slower than GNN inference. Fortunately, this situation has changed recently - along with the approaching death of Moore's law, computer architects are inventing new types of computers with diverse computing powers (other than digital processors). Among these computers, a computer that can swiftly solve Ising models at low cost (nanoseconds and milliwatts -- 10^6 faster than being solved on digital computers) has recently emerged. The computer is essentially a CMOS-based Ising machine, which benefits from mature semiconductor manufacturing processes (guaranteeing its easy manufacturing and integration) and attracts much attention to the Ising model in recent 2 years. It has already enabled a wide discussion of using methods rooted from the Ising model and the Ising machine to solve different types of problems. Despite great promise and increasing attention, this type of research is still in its early stage, like quantum computing, waiting for its killer applications, while unlike quantum computing, holding short-term promise of unleashing the power of nature in the real world. In the current stage of the development of this method, the binary limitation is believed to be one of the most important challenges to this adoption in the real world. The essence of this challenge is that the vanilla Ising formulation is already complex enough to be realized and implemented as a CMOS chip – any augmentation to the Ising formulation/Hamiltonian that complicates the circuit implementation should be very cautious. This paper tries to break this limitation by finding a new Ising formulation with slight augmentation that both enables real-value support and avoids complicating the Ising hardware.
>
>
> **Comment 3**. From linear to quadratic.\
> **Response:**
>
> As detailed above, the biggest challenge in extending potentials of Ising method from binary to real value is that the Hamiltonian formulation needs to remain simple, such that the Ising machine can be realistically manufactured – note that the Ising machine to support vanilla is already very challenging. To this end, our design goal is to make the change as slight as possible into the Hamiltonian while enabling the support of real value (enabling non-polarized voltages in the machine).

---

> > ### Author Response · Authors · 2023-11-22
> >
> > As the end of the discussion session approaches, we would like to take the last chance to sincerely thank you for all the constructive and insightful reviews and suggestions. Any comments on our responses and revision are very much appreciated! Also, thank you in advance for your continued efforts in the rest of the review process. And if you are celebrating it, Happy Thanksgiving!

---

### Author Response · Authors · 2023-11-19
**General Comments**

**Summary:** NP-GL proposes a new physics-rooted graph learning method that effectively transfers real-world graph learning problems into the natural process of physical dynamic systems composed of CMOS components. With this transfer, the intrinsic computational power rooted in nature within the dynamic systems (automatically evolving towards and chasing the more stable states with lower energy driven by law of entropy increase) is enabled in advancing graph learning. Results demonstrate that such nature-powered learning method can outperform GNNs in various important real-world graph learning problems.

**Significance of proposed NP-GL:** To the best of our knowledge, NP-GL is the first effort made to successfully transfer real-world learning problems into a natural process. With the success of this transfer, in NP-GL, the inherent and huge computational power of nature in dynamic systems (automatically chasing stable states with lower energy) is unleashed in learning problems, leading to significant improvement in efficiency. It is worthy to mention that exploring computing power other than digital computers is becoming increasingly important due to the approaching end of Moore’s law, which leads to the slowdown of the advancement of digital processor power. Moreover, NP-GL is not just a theoretical concept. Instead, a dynamic system with CMOS components (which is being taped out recently) is proposed. Therefore, different from most efforts on Ising models, NP-GL can be evaluated end-to-end with respect to accuracy, latency, and energy cost with real-world applications, and can be compared with other methods.

**Why graph learning and versatility:** The manuscript focuses on graph learning, as we observe that many real-world graph learning problems are actually dynamic systems from the perspective of physics (e.g. power grid, social media, traffic, air pollution). Inspired by this fact, we are encouraged to explore whether a method (NP-GL) using a dynamic system and its inherent nature power to solve a dynamic system problem can be a more efficient solution than the ones relying on digital methods such as GNNs plus GPUs. A good analogy is quantum computing: It is clearly more efficient and accurate to use a quantum computer to understand quantum effects than methods based on digital solutions. Note that this method does not only work with graph problems. Based on our preliminary evaluation, it can also do 3D synthesis like NeRF and image generation like stable diffusion. However, those researches are still in their early stage, and the current quality of the nature-powered solutions to those vision tasks is so far not as impressive as the one in graph learning. Moreover, it is worth mentioning that the NP-GL method is not an alternative GL method for GNNs; instead, it can be considered as a complementary method to GNNs. Due to its low latency and energy cost with good accuracy, it can be used to drastically reduce GNN workload (e.g. GNN only operates on a small subgraph while NP-GL method is in charge of the rest). Furthermore, it can also be seen as a nature-powered generative AI method with negligible cost, holding the potential to fundamentally reduce the computational cost of deep learning in the future (if interested, more details in Reviewer w2U6 - Question 3).

---

### Meta-Review · Area_Chair_DLpG · 2023-12-23

**Metareview:**

The paper introduces a novel Nature-Powered Graph Learning (NP-GL) framework, extending nature-powered computation from binary-valued problems to real-valued problems. The paper designed a hardware-friendly Hamiltonian and a new nature-based computer. The proposed method is shown to exhibit significant speedup and energy savings over graph neural nets on a few datasets. The authors also discussed the potential of the proposed method on applications beyond GNN, such as data compression and 3D synthesis. The reviewers are mostly supportive of this paper. I recommend acceptance.

**Justification For Why Not Higher Score:**

While the proposed method works quite well in six datasets, these datasets are relatively small, and the power of the proposed method in large-scale datasets is yet to be studified.

**Justification For Why Not Lower Score:**

The paper proposes a novel approach that has the potential in increasing the efficiency and power assumption compared to existing methods.

---

### Decision · Program_Chairs · 2024-01-16

Accept (poster)